# The Added Value of Radiographs in Diagnosing Knee Osteoarthritis Is Similar for General Practitioners and Secondary Care Physicians; Data from the CHECK Early Osteoarthritis Cohort

**DOI:** 10.3390/jcm9103374

**Published:** 2020-10-21

**Authors:** Qiuke Wang, Jos Runhaar, Margreet Kloppenburg, Maarten Boers, Johannes W. J. Bijlsma, Sita M. A. Bierma-Zeinstra

**Affiliations:** 1Department of General Practice, Erasmus MC University Medical Center Rotterdam, 3015 GD Rotterdam, The Netherlands; q.wang@erasmusmc.nl (Q.W.); s.bierma-zeinstra@erasmusmc.nl (S.M.A.B.-Z.); 2Department of Rheumatology, Leiden University Medical Center, 2333 ZA Leiden, The Netherlands; G.Kloppenburg@lumc.nl; 3Department of Epidemiology & Data Science, Amsterdam Rheumatology & Immunology Center, Amsterdam UMC, Vrije Universiteit Amsterdam, 1081 HV Amsterdam, The Netherlands; eds@amsterdamumc.nl; 4Department of Rheumatology and Clinical Immunology, University Medical Centre Utrecht, 3584 CX Utrecht, The Netherlands; j.w.j.bijlsma@umcutrecht.nl; 5Department of Orthopaedics, Erasmus MC University Medical Center Rotterdam, 3015 GD Rotterdam, The Netherlands

**Keywords:** knee osteoarthritis, radiography, general practitioner, secondary care physician, diagnosis

## Abstract

Objective: The purpose of this study was to evaluate the added value of radiographs for diagnosing knee osteoarthritis (KOA) by general practitioners (GPs) and secondary care physicians (SPs). Methods: Seventeen GPs and nineteen SPs were recruited to evaluate 1185 knees from the CHECK cohort (presenters with knee pain in primary care) for the presence of clinically relevant osteoarthritis (OA) during follow-up. Experts were required to make diagnoses independently, first based on clinical data only and then on clinical plus radiographic data, and to provide certainty scores (ranging from 1 to 100, where 1 was “certainly no OA” and 100 was “certainly OA”). Next, experts held consensus meetings to agree on the final diagnosis. With the final diagnosis as gold standard, diagnostic indicators were calculated (sensitivity, specificity, positive/negative predictive value, accuracy and positive/negative likelihood ratio) for all knees, as well as for clinically “certain” and “uncertain” knees, respectively. Student paired *t*-tests compared certainty scores. Results: Most diagnoses of GPs (86%) and SPs (82%) were “consistent” after assessment of radiographic data. Diagnostic indicators improved similarly for GPs and SPs after evaluating the radiographic data, but only improved relevantly in clinically “uncertain” knees. Radiographs added some certainty to “consistent” OA knees (GP 69 vs. 72, *p* < 0.001; SP 70 vs. 77, *p* < 0.001), but not to the consistent no OA knees (GP 21 vs. 22, *p* = 0.16; SP 20 vs. 21, *p* = 0.04). Conclusions: The added value of radiographs is similar for GP and SP, in terms of diagnostic accuracy and certainty. Radiographs appear to be redundant when clinicians are certain of their clinical diagnosis.

## 1. Introduction

In routine clinical practice, the diagnosis of knee osteoarthritis (KOA) is usually made based on the clinician’s expertise, and radiographs are frequently used to confirm clinical suspicion of KOA [1,2]. However, there are insufficient data on the necessity and the potential role of radiographs in the diagnostic process.

The European League Against Rheumatism Recommendations (EULAR) reported that three symptoms (knee pain, morning stiffness less than 30 min and functional limitation) combined with three clinical signs (crepitus, restricted range of motion and bone enlargement) could predict 99% radiographic KOA [3]. Similarly, recent studies showed that clinical manifestations, such as knee pain, crepitus, joint line tenderness, bony swelling and pain on flexion/extension could be used for identifying radiographic KOA [4,5,6]. Current recommendations advise not to use imaging in patients with typical OA presentations, but these were mainly based on expert opinion [7,8].

As a common and chronic disease [8,9,10], KOA is usually diagnosed both by general practitioners (GPs) and secondary care physicians (SPs). The added diagnostic value of radiographs can be different between the two kinds of clinicians given their different clinical expertise. However, there is no scientific literature on this aspect.

In this study, we recruited both GPs and SPs with osteoarthritis (OA) expertise to assess clinical vignettes taken from the CHECK cohort study (a longitudinal cohort study of primary care patients with knee complaints suggestive for early stage KOA, followed for 10 years) of potential KOA patients and to provide diagnoses based on either clinical data alone, or clinical combined with radiographic data. The aim of this study was to evaluate the added value of radiographs above clinical findings in diagnosing KOA and to see whether this differed between GPs and SPs.

## 2. Methods

### 2.1. Clinical Experts

The protocol has been approved by the Ethical Committee of UMC Utrecht (protocol number 02/017-E). We recruited experts who fulfilled one of the following criteria in this study: (i) had a degree in general practice, orthopedics, rheumatology or sports medicine for 2 or more years; (ii) were in-training for a degree in these specialties combined with a PhD in OA research.

We tested experts’ characteristics by querying them on the number of OA patients treated per week, experience in OA treatment (years), and their perception on the importance of radiographs in making the diagnosis.

### 2.2. Clinical and Radiographic Data

For the present study, we included all patients from the CHECK cohort [11,12,13]. The CHECK cohort recruited patients between October 2002 and September 2005, and all patients were supposed to be followed for 10 years. Patients whose medical records and radiographs were available from 5 up to 10-year follow-up were included in this study.

Clinical data, obtained at a 5, 8 and 10-year follow up, consisted of demographics (including sex, age, BMI (body mass index), racial background, marital status, menopausal status, educational level, chronic diseases, occupation, smoking status and alcohol usage), physical examinations (presence of knee pain, morning stiffness in knee, knee warmth, bony tenderness, crepitus, knee pain on extension and flexion, range of motion (extension and flexion)), WOMAC (Western Ontario and McMaster Universities Osteoarthritis Index) subscales of pain, function, and stiffness, NRS (numeric rating scale) pain scores and incidence of other diseases (quadriceps tendinitis, intra-articular fracture, Baker’s cyst, ligament or meniscus damage, osteochondritis dissecans, plica syndrome and septic arthritis).

Radiographic data consisted of scores from centralized reading by trained readers evaluating standard weight bearing (posterior-anterior fixed flexion view) knee radiographic films at 5, 8, and 10 years follow up (for details see [12]). The scores included information on tibial attrition, femoral/tibial sclerosis, joint space narrowing, femoral/tibial osteophytes and Kellgren and Lawrence grades. Both posterior-anterior fixed flexion and lateral films were also made available to the experts.

Appendix A summarizes all clinical and radiographic data presented to the experts. All data were stored and presented in special software (built in-house) for optimal presentation. The software recorded actual access to the radiographic films.

### 2.3. Obtaining Diagnoses

Before starting the diagnostic process, all experts received written information and completed two example patients to get familiarized with the procedures and software. We obtained expert diagnosis between June 2018 and January 2019.

Experts were divided into pairs; each pair consisted of one GP and one SP, where possible. The diagnostic process is presented in Figure 1. Each pair assessed the same subset of knees (40–50 patients). First, longitudinal clinical data of each patient were presented. Each expert evaluated these independently and, for each knee, chose between “yes, clinically relevant OA has developed” and “no, clinically relevant OA has not developed”. In addition, the experts had to provide their certainty on a 1 to 100 scale (integers with unit of 1), where 1 was “certainly no OA” and 100 was “certainly OA”. If a knee was diagnosed as “OA”, the certainty score had to be between 51 and 100, with a higher score expressing greater certainty; if the knee was diagnosed as “no OA”, certainty score had to be between 1 and 49, with a lower score expressing greater certainty. Next, access to longitudinal radiographic data and films was activated. Experts were asked the same questions and had to provide new certainty scores. At this stage, experts had read-only access to the clinical data and their corresponding diagnoses.

After individually finishing all these evaluations, knees assigned certainty scores >30 and <70 were defined as “uncertain”, the remainder as “certain”. Where the two experts agreed (yes/no OA, regardless of certainty) the diagnosis automatically became final. Each pair held a consensus meeting to re-assess the knees where the individual diagnoses were discrepant, except if both experts were “uncertain”. At that meeting the expert pair evaluated both clinical and radiographic data of the discrepancies, as done when evaluating these individually, and made a final diagnosis together. Knees where no consensus could be reached and those for which the experts disagreed after the individual scoring, but both were “uncertain”, were all labeled as “consensus based uncertain”.

### 2.4. Statistics

Categorical variables were presented as counts and percentages and normally distributed continuous data as mean ± standard deviation. Experts characteristics were compared by Mann–Whitney U test or Wilcoxon W test, where appropriate. The number of consistent and amended diagnoses after assessment of the radiographic data were presented for GP and SP, split for no OA and OA diagnoses obtained when evaluating the clinical data only. Chi-square tests were used for comparing diagnoses before and after viewing radiographic data in GP/SP. We calculated sensitivity, specificity, positive/negative predictive value (PPV/NPV), accuracy, positive/negative likelihood ratio (LR+/–) and their 95% CIs (confidence intervals) for the GP and SP diagnosis separately, with the consensus based final diagnosis as gold standard. Next, we split all knees into clinically “certain” (individual certainty scores ≤30 or ≥70, based on clinical data only) and clinically “uncertain” (individual certainty scores >30 and <70, based on clinical data only) and the same diagnostic indicators were calculated within both groups. The primary objective of the present study was to assess the clinically relevant value of radiographs. With the consideration that statistical tests could be too sensitive to detect minor differences in such a large sample, we did not apply statistical analysis for comparing the above described diagnostic indicators. As no comparable results have been reported before, outcomes were deemed exploratory.

For the analysis of certainty scores, the knees were divided into four subgroups: “consistent OA” (the clinical diagnosis of OA was retained after viewing radiographic data), “amended to no OA” (clinical diagnosis OA amended to no OA after viewing radiographic data); and likewise, “consistent no OA”, and “amended to OA”. Paired *t*-tests assessed whether diagnostic certainty was improved with radiographic information, in “consistent OA” and “consistent no OA” knees. To assure robustness of the results, a sensitivity analysis compared certainty scores between left knees only, with a paired *t*-test. 

Analysis was performed with SPSS version 25.0 (IBM, Chicago, IL, USA); the significance level was 0.05 using a 2-sided *p* value for all tests.

## 3. Results

### 3.1. Experts and Patients

A total of 36 experts were recruited, 17 GPs and 19 SPs (8 orthopedists, 9 rheumatologists and 2 sports physicians). Seventeen pairs of one GP and one SP were formed, and the remaining pair included one orthopedist and one rheumatologist. 

Expert characteristics are shown in Table 1. Among all the characteristics, only the perceived importance of radiography was significantly different between GP and SP (*p* < 0.001).

The study included 761 patients with 1185 symptomatic knees; 79% female, mean (SD) age 56 (5) years. The pairs formed by one GP and one SP evaluated 1106 knees, and the pair consisted of two SP did for 79 knees. During the diagnostic process, GP viewed the actual films in 45% of the knees and SP did this in 69% of the knees. 

### 3.2. Diagnostic Abilities

#### 3.2.1. General Practitioners

GPs diagnosed 42% of knees as OA based on the clinical data only and 44% after viewing radiographic data. In total, 86% of diagnoses were consistent after viewing radiographic data; 6% OA knees were amended to no OA, and 8% no OA knees to OA (Figure 2). Of the 14% amended diagnoses, 8% were deemed correct, compared to the final diagnosis (Table 2). In general, the changes in diagnoses were statistically significant (*p* < 0.001) and all diagnostic indicators were somewhat improved after viewing radiographic data (Table 3).

GPs were uncertain about 41% of their clinical diagnoses. They were much more likely to amend uncertain diagnoses than their certain diagnoses (23% uncertain vs. 7% of certain diagnoses). Furthermore, the rate of correct amendments in clinically uncertain diagnoses was 2 times higher than that in certain diagnoses (Table 4). Without radiographic assessment, the GP diagnostic ability was good in certain diagnoses (PPV = 0.82 (95% CI, 0.76–0.87), NPV = 0.71 (95%CI, 0.67–0.75)), but poor in uncertain diagnoses (PPV = 0.59 (95% CI, 0.53–0.65), NPV = 0.57 (95%CI, 0.43–0.58)). Diagnostic abilities were much improved by radiographs in uncertain diagnoses (Table 5). In clinically certain diagnoses, slight improvements were found in PPV, NPV and LR− (PPV: 0.82 to 0.84; NPV: 0.71 to 0.77; LR−: 0.30 to 0.18), LR+ values were consistently greater than 10 (15.39 to 31.69); in clinically uncertain diagnoses, all indicators were much improved (PPV: 0.59 to 0.73; NPV: 0.51 to 0.61; LR+: 2.14 to 4.89; LR−: 0.30 to 0.04). 

#### 3.2.2. Secondary Care Physicians

SPs diagnosed 39% of knees as OA based on clinical data only and 49% after viewing radiographic data. In total, 82% of diagnoses were consistent after viewing radiographic data; 4% OA knees were amended to no OA and 14% no OA knees to OA (Figure 2). Of the 18% amended diagnoses, 9% were deemed correct as compared with the final diagnosis (Table 2). In general, the changes in diagnoses were statistically significant (*p* < 0.001) and all diagnostic indicators were somewhat improved after viewing radiographic data (Table 3). 

SPs were uncertain about 36% of their clinical diagnoses. They were much more likely to amend uncertain diagnoses than certain diagnoses (27% uncertain vs. 14% of certain diagnoses). Furthermore, the rate of correct amendments in clinically uncertain diagnoses was 3 times higher than that in certain diagnoses (Table 4). Without radiographic assessment, the SP diagnostic ability was good in certain diagnoses (PPV = 0.78 (95%CI, 0.72–0.83), NPV = 0.74 (95%CI, 0.70–0.77)), but poor in uncertain diagnoses (PPV = 0.57 (95%CI, 0.50–0.63), NPV = 0.42 (95%CI, 0.35–0.49)). Diagnostic abilities were much improved by radiographs in uncertain diagnoses (Table 5). In clinically certain diagnoses, some fluctuations were found in PPV, NPV and LR− (PPV: 0.78 to 0.71; NPV: 0.74 to 0.86; LR−: 0.28 to 0.11), LR+ values were consistently greater than 10 (14.98 to 21.02); in clinically uncertain diagnoses, all indicators were much improved (PPV: 0.57 to 0.66; NPV: 0.42 to 0.68; LR+: 1.88 to 4.81; LR−: 0.45 to 0.05).

### 3.3. Diagnostic Certainties

Diagnostic certainty scores are presented in Figure 3. For GP, diagnostic certainty improved somewhat in the “consistent OA” knees (69 ± 12 vs. 72 ± 14, *p* < 0.001), while no significant improvement was found in “consistent no OA” knees (21 ± 13 vs. 22 ± 14, *p* = 0.16). Sensitivity analysis showed very similar results (see Appendix A). 

For SPs, diagnostics certainty improved somewhat in the “consistent OA’” knees (70 ± 12 vs. 77 ± 15, *p* < 0.001). Diagnostic certainty of “consistent no OA” was minimally but significantly altered after viewing radiographic data (SP 20 ± 13 vs. 21 ± 15, *p* = 0.04). Sensitivity analysis showed very similar results (see Appendix A).

## 4. Discussion

In this study, we showed that radiographs added only to the diagnostic ability of both GPs and SPs in clinically “uncertain” diagnoses. Overall, diagnostic ability, diagnostic certainty and the added value of radiographs were very similar for GP and SP.

Both GP and SP amended some of their diagnoses after viewing the radiographic data, but the majority of diagnoses remained the same. As a time-consuming, costly and potentially radiation-hazardous examination, radiographs seem to be redundant in most cases suspicious for knee OA. The diagnostic abilities of GP and SP, without access to radiographic data, were already comparable to findings in other chronic musculoskeletal diseases, such as lumbar spinal stenosis [14] and lumbar disc herniation [15]. Therefore, for clinically “certain” knees, diagnostic abilities based on clinical data only should be considered as good enough, in contrast to clinically “uncertain” knees. Our results support expert recommendations and the results of previous studies [3,4,6,7,8], where diagnoses based on clinical findings were found to be reliable and where radiographs were deemed unnecessary for diagnosing typical KOA.

On the other hand, after viewing radiographic data, diagnostic indicators for both GP and SP were much improved in clinically “uncertain” knees. Likelihood ratios, calculated by using sensitivity and specificity, can directly reflect the ability of diagnosing OA/no OA [16,17]. LR+ is deemed clinically meaningful if greater than 10 and LR− when lower than 0.1. In this study, radiographs helped to improve LR+ from 2 to 5 in “uncertain” knees. According to a previous literature report [17], it indicates the probability of correct diagnosis for OA knees was increased by 15%. LR‒ was improved from 0.4 to 0.05. The probability of a correct diagnosis of no OA knees was increased by 25%. Hence, we believe the improvements in “uncertain” knees are clinically meaningful, and radiographs could be considered in these cases.

Both GPs and SPs seemed to be more certain of their radiographically confirmed OA (“consistent OA”) diagnoses. However, as other joint diseases were excluded from the CHECK cohort at baseline [11,13] and the incidence of these diseases during follow-up was quite low (3%), all the abnormalities presented in radiographic data would direct experts to an OA diagnosis, rather than to other conditions. In other words, our results could be inflated compared to real practice. Furthermore, because this is the first study of its kind, it remains unclear whether the certainty improvements are clinically relevant. On the other hand, our results did not support the strategy of using radiographs for improving certainty of no OA diagnoses. On average, the experts were already “fairly” certain (certainty scores < 30) about clinically no OA diagnoses and neither GPs nor SPs became more certain after viewing radiographic data in consistent no OA knees.

In this study, we provided standardized radiographic scores to the experts, which should be helpful to diminish the bias of image reading skill differences between different experts. Even though actual films were also available if required, not all films were viewed by experts. SP seemed to be more interested in the actual films than GP in this study. This aligns with their characteristics and also can be explained by their differences in image interpretation skills, which is correlated with image exposure in daily clinical work [18,19].

Since the major aim of this study was to evaluate the added value of radiographs above clinical findings in diagnosing KOA, we did not perform specific statistical analysis on the diagnostic results between GPs and SPs. Generally, SPs amended slightly more diagnoses than GPs after viewing radiographic data (18% vs. 14%), which could be explained by the expert characteristics as SPs place more emphasis on radiographs, but the rate of correct amendment was similar (9% vs. 8%). Furthermore, there was no obvious difference among diagnostic indicators between GPs and SPs either before or after viewing radiographic data. Similar results were also found in certainty scores. Therefore, we believe the added value of radiographs should be considered as similar for GPs and SPs.

This study has limitations. First, there is likely some incorporation bias when we compare the GPs’ and SPs’ diagnoses to the consensus-based final diagnoses, because the individual diagnoses which both experts agreed on were incorporated into final diagnosis automatically [20]. That means the absolute values of diagnostic indicators in these comparisons are potentially overestimated. Decary et al. reported that the amount of overestimation of sensitivity and specificity caused by incorporation bias depended on the true specificity of the test method [21]. It was impossible to quantify the overestimation in the current study, due to the lack of the true specificity of expert diagnoses. A second potential concern, 424 patients with bilateral knee complaints were included in this study. Two knees from the same patient shared the same demographic data and WOMAC scales. In principle, it is inappropriate to view them as fully independent observations. However, our sensitivity analysis limited to left knee data only yielded results similar to the main analysis, suggesting this is not a problem in our dataset. Third, standard radiographic scores (i.e., Kellgren and Lawrence grade) as well as actual films were provided to experts in this study, which differs from the scenario of routine clinical work. Even so, most clinical diagnoses remained same after viewing both the scores and films. This, to some degree, supports our conclusion that radiographs seem to be redundant in most cases. Fourth, we did not provide the skyline view to the experts, so some patellofemoral joint OA might be missed. However, we believe any influence on our conclusions is limited because the prevalence of patellofemoral joint OA in the CHECK cohort was quite low (4.6%) [22], and its presence would also have been suggested by the lateral radiograph findings as well as clinical history and physical examination, e.g., knee crepitus [23]. Fifth, the process of obtaining final diagnosis could have been influenced by authority, since SPs likely have more authority than GPs. In this case, diagnostic indicators of SPs would be higher than those of GPs. As the diagnostic indicators were quite similar between GPs and SPs, we believe this is not a big issue.

In conclusion, radiography could be of importance in cases where the clinical diagnosis of KOA is uncertain. Radiographs helped to improve the certainty of OA diagnoses, but the clinical relevance of this improvement is unclear. Overall, all results were similar for GPs and SPs.

## Figures and Tables

**Figure 1 jcm-09-03374-f001:**
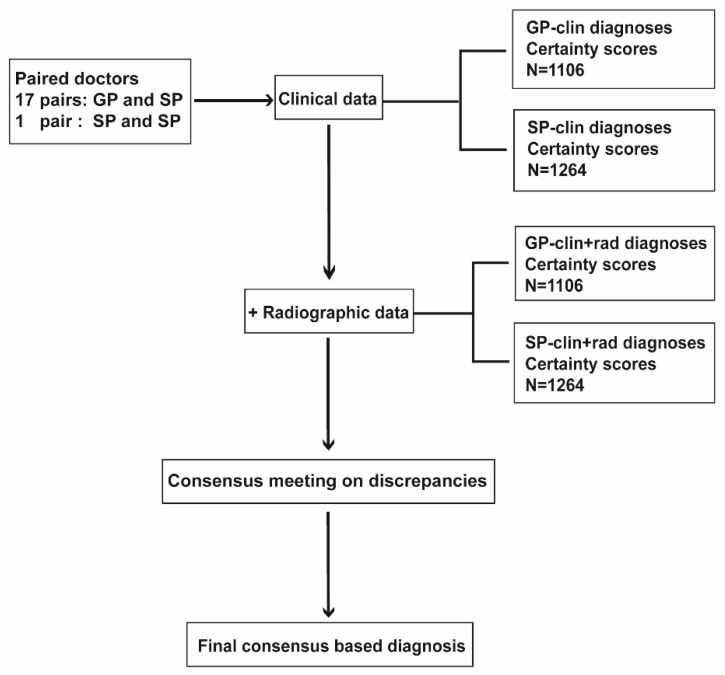
Diagnostic process. OA, osteoarthritis; GP, general practitioner; SP, secondary care physician; Clin diagnosis, diagnoses based on clinical data only; clin + rad, diagnoses based on clinical and radiographic data.

**Figure 2 jcm-09-03374-f002:**
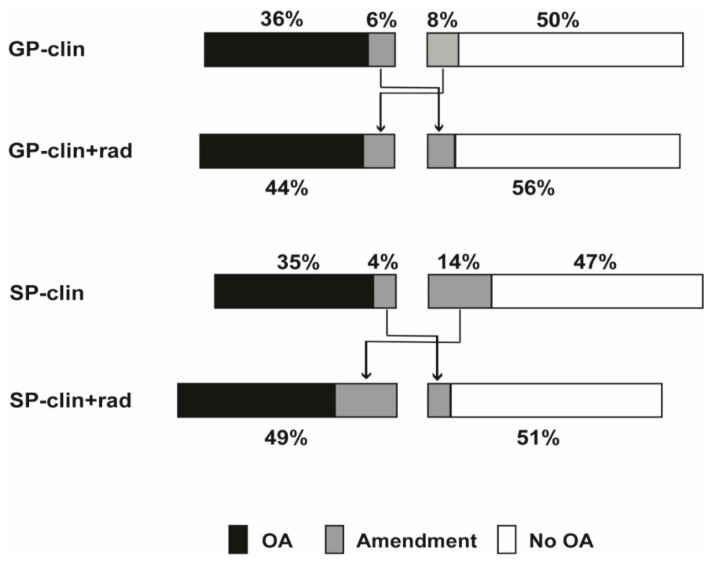
General practitioner (GP) and secondary care physician (SP) diagnoses based on clinical data (clin), clinical combined with radiographic data (clin + rad); OA, osteoarthriti.

**Figure 3 jcm-09-03374-f003:**
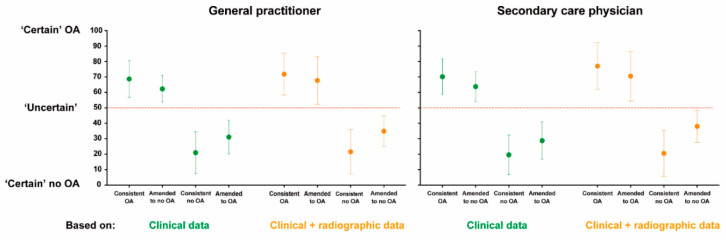
Certainty scores of four subgroups before and after viewing radiographic data (means and standard deviations).

**Table 1 jcm-09-03374-t001:** Expert characteristics.

	General Practitioner (N = 17)	Orthopedist (N = 8)	Rheumatologist (N = 9)	Sports Physician (N = 2)	*p* Values ^&^
Importance radiograph *, median (range)	2 (1–4)	4 (4–4)	3 (2–4)	3 (3–3)	<0.001
Number of OA patients treated per week, mean (SD)	5 (3)	53 (25)	6 (7)	3 (2)	0.06
Years of experience, mean (SD)	12 (9)	14 (5)	17 (9)	20 (14)	0.10

OA, osteoarthritis; * Perceived importance of radiography for making the diagnosis of knee osteoarthritis: 1, not important; 2, of minor importance; 3, somewhat important; 4, very important; ^&^ Comparing characteristics between GP and SP.

**Table 2 jcm-09-03374-t002:** Expert diagnoses and certainty scores.

	OA %	Certainty Scores of OA Knees Mean (SD)	No OA %	Certainty Scores of No-OA Knees Mean (SD)	Consistent Diagnoses * %	Amended Diagnoses %	Correctly Amended ^#^ %
**GP-clin, N = 1106**	42 ^&^	68 (12)	58 ^&^	22 (14)	86	14	8
**GP-clin + rad, N = 1106**	44 ^&^	71 (14)	56 ^&^	23 (15)			
**SP-clin, N = 1264**	39 ^$^	69 (12)	61 ^$^	22 (13)	82	18	9
**SP-clin + rad, N = 1264**	49 ^$^	75 (16)	51 ^$^	22 (15)			

OA, osteoarthritis; GP-clin, general practitioners’ diagnoses based on clinical data only; GP-clin + rad, general practitioners’ diagnoses based on clinical and radiographic data; SP-clin, secondary care physicians’ diagnoses based on clinical data only; SP-clin + rad, secondary care physicians’ diagnoses based on clinical and radiographic data; * the clinical diagnosis of OA/no OA was retained after viewing radiographic data; ^#^ diagnoses correctly amended after viewing radiographs, compared to final diagnosis; ^&^
*p* <0.001, chi-square test for comparing the diagnoses before and after viewing radiographs in GP; ^$^
*p* <0.001, chi-square test for comparing the diagnoses before and after viewing radiographs in SP.

**Table 3 jcm-09-03374-t003:** Comparison of initial and consensus-based final diagnosis.

		Final Diagnosis			Sensitivity (95%CI)	Specificity (95%CI)	PPV (95%CI)	NPV (95%CI)	Accuracy (95%CI)	LR+ (95%CI)	LR− (95%CI)
		OA *	No A *	Uncertain *^,†^							
GP-clin (N = 1106)	OA	29	6	7	0.76 (0.72–0.80)	0.86 (0.82–0.89)	0.68 (0.64–0.73)	0.66 (0.62–0.69)	0.67 (0.63–0.71)	5.23 (4.24–6.63)	0.28 (0.24–0.33)
	No OA	9	38	11							
GP-clin + rad (N = 1106)	OA	34	3	6	0.89 (0.86–0.92)	0.92 (0.90–0.95)	0.78 (0.74–0.82)	0.72 (0.68–0.75)	0.75 (0.71–0.78)	11.75 (8.60–16.04)	0.12 (0.09–0.16)
	No OA	4	41	12							
SP-clin (N = 1264)	OA	26	6	7	0.73 (0.69–0.77)	0.88 (0.84–0.90)	0.67 (0.62–0.71)	0.66 (0.62–0.69)	0.66 (0.62–0.70)	5.86 (4.68–7.32)	0.31 (0.26–0.36)
	No OA	10	40	11							
SP-clin + rad (N = 1264)	OA	33	4	12	0.93 (0.90–0.95)	0.92 (0.90–0.94)	0.69 (0.65–0.72)	0.82 (0.78–0.85)	0.75 (0.71–0.78)	11.84 (8.94–15.70)	0.08 (0.06–0.11)
	No OA	3	42	6							

OA, osteoarthritis; GP-clin, general practitioners’ diagnoses based on clinical data only; GP-clin + rad, general practitioners’ diagnoses based on clinical and radiographic data; SP-clin, secondary care physicians’ diagnoses based on clinical data only; SP-clin + rad, secondary care physicians’ diagnoses based on clinical and radiographic data; PPV, positive predictive value; NPV, negative predictive value; LR+, positive likelihood ratio; LR−, negative likelihood ratio. The diagnostic indicators were calculated by treating the final diagnosis as reference standard and using all knees, including OA, no OA and uncertain diagnoses, in the calculations; * Percentages (%) of patients of each subgroup; ^†^ Knees diagnosed as “consensus based uncertain”.

**Table 4 jcm-09-03374-t004:** Final diagnoses and rate of amendment split by certainty of initial clinical diagnosis (% of patients of each subgroup).

		Clin+ Rad Clin	OA	No OA	Consistent Diagnoses *	Amended Diagnoses	Correctly Amended ^#^
Clinically ‘certain’ knees	GP (N = 658)	OA	28	2	93	7	5
		no OA	5	65			
	SP (N = 811)	OA	27	2	86	14	5
		no OA	12	59			
Clinically ‘uncertain’ knees	GP (N = 448)	OA	48	13	77	23	13
		no OA	10	29			
	SP (N = 453)	OA	49	9	73	27	16
		no OA	18	24			

OA, osteoarthritis; Clin: knees diagnosed as OA or no OA based on clinical data only; Clin + rad: knees diagnosed as OA or no OA based on clinical and radiographic data; * the clinical diagnosis of OA/no OA was retained after viewing radiographic data; ^#^ diagnoses correctly amended after viewing radiographs, compared to the final diagnosis.

**Table 5 jcm-09-03374-t005:** Comparing experts’ clinically certain/uncertain diagnoses to consensus based final diagnosis.

			Final Diagnosis									
			OA *	No OA *	Uncertain *^,†^	Sensitivity (95%CI)	Specificity (95%CI)	PPV (95%CI)	NPV (95%CI)	Accuracy (95%CI)	LR+ (95%CI)	LR− (95%CI)
Clinically ‘certain’ knees	GP-clin (N = 658)	OA	24	2	3	0.71 (0.65–0.77)	0.95 (0.92–0.97)	0.82 (0.76–0.87)	0.71 (0.67–0.75)	0.74 (0.71–0.81)	15.39 (9.47–25.02)	0.30 (0.25–0.37)
		No OA	10	50	11							
	GP-clin + rad (N = 658)	OA	28	1	4	0.82 (0.77–0.87)	0.97 (0.95–0.99)	0.84 (0.78–0.88)	0.77 (0.72–0.80)	0.79 (0.75–0.84)	31.69 (16.58–60.56)	0.18 (0.14–0.24)
		No OA	6	51	10							
	SP-clin (N = 811)	OA	23	3	4	0.74 (0.68–0.79)	0.95 (0.93–0.97)	0.78 (0.72–0.83)	0.74 (0.70–0.77)	0.75 (0.71–0.80)	14.98 (9.90–22.67)	0.28 (0.22–0.34)
		No OA	8	52	10							
	SP-clin + rad (N = 811)	OA	27	2	9	0.90 (0.85–0.93)	0.96 (0.93–0.97)	0.71 (0.66–0.76)	0.86 (0.82–0.89)	0.80 (0.74–0.84)	21.02 (13.51–32.71)	0.11 (0.08–0.16)
		No OA	3	53	6							
Clinically ‘uncertain’ knees	GP-clin (N = 448)	OA	36	11	13	0.81 (0.75–0.86)	0.62 (0.53–0.70)	0.59 (0.53–0.65)	0.51 (0.43–0.58)	0.56 (0.48–0.62)	2.14 (1.72–2.67)	0.30 (0.22–0.40)
		No OA	9	20	11							
	GP-clin + rad (N = 448)	OA	43	6	9	0.97 (0.93–0.98)	0.80 (0.73–0.86)	0.73 (0.67–0.78)	0.61 (0.54–0.68)	0.68 (0.60–0.73)	4.89 (3.51–6.83)	0.04 (0.02–0.09)
		No OA	2	25	15							
	SP-clin (N = 453)	OA	33	11	14	0.72 (0.66–0.78)	0.62 (0.53–0.70)	0.57 (0.50–0.63)	0.42 (0.35–0.49)	0.51 (0.42–0.56)	1.88 (1.49–2.37)	0.45 (0.36–0.57)
		No OA	13	18	11							
	SP-clin + rad (N = 453)	OA	44	6	17	0.96 (0.92–0.98)	0.80 (0.72–0.86)	0.66 (0.60–0.71)	0.68 (0.60–0.76)	0.67 (0.60–0.73)	4.81 (3.40–6.79)	0.05 (0.02–0.10)
		No OA	2	23	8							

OA, osteoarthritis; GP-clin, general practitioners’ diagnoses based on clinical data only; GP-clin + rad, general practitioners’ diagnoses based on clinical and radiographic data; SP-clin, secondary care physicians’ diagnoses based on clinical data only; SP-clin + rad, secondary care physicians’ diagnoses based on clinical and radiographic data; PPV, positive predictive value; NPV, negative predictive value; LR+, positive likelihood ratio; LR−, negative likelihood ratio. The diagnostic indicators were calculated by treating the final diagnosis as reference standard and using all knees, including OA, no OA and uncertain diagnoses, in the calculations; * Percentages (%) of patients of each subgroup; ^†^ Knees diagnosed as “consensus based uncertain”.

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
