# Peer review of "The Added Value of Radiographs in Diagnosing Knee Osteoarthritis Is Similar for General Practitioners and Secondary Care Physicians; Data from the CHECK Early Osteoarthritis Cohort"

_jcm, 2020, doi:10.3390/jcm9103374_

Round 1
Reviewer 1 Report
I had the opportunity to review the article entitled: ‘The added value of radiographs in diagnosing knee osteoarthritis is similar for general practitioners and secondary care physicians; data from the CHECK early osteoarthritis cohort.’ that is submitted for publication.
Comments: This study evaluated the value of radiographic data in the diagnosis of knee osteoarthritis (OA) by general physician (GP) and secondary care physician (SP). Although this study provides new information that has not been reported so far, it has some concerns. The core information of knee OA is missing from radiographic analysis. Moreover, the statistical analysis is very poor. Therefore, it will not be able to accept this manuscript for publication in its present form. However, I think that this paper has sufficient potential to be reconsidered and evaluated again by reviewers and editors if the authors significantly alter and rework the manuscript.
Abstract
- Line 21: ‘Seventeen’… The beginning of the sentence should not start with a number.
- Line 21, 55: CHECK… The authors should define the abbreviations at their first appearance. Although it is widely accepted.
- Hypothesis?
Methods
- IRB approval?
- Study period?
- Line 74: In general, BMI is also included in the demographics. Is there any reason to separate this?
- Line 84-85: This is the major concern of this study. Did the authors only evaluate the standing AP/LAT view? It is very important to evaluate the posterior femoro-tibial joint or patello-femoral joint in radiographic diagnosis of the knee OA. Didn't the authors assess the Rosenberg view or the Merchant view? I think this must be evaluated.
- Line 96-102: Is the score in units of 1 point?
- Statistics: Which statistical program did the authors use? What was the significant ranges of p-value?
- Line 119: ‘Ssensitivity’
Results
- To evaluate radiographic accuracy between GP and SP, it can be interesting to compare the evaluation results for K-L grade. Please also add the judgment result of K-L grade between each clinician to the result.
- Line 137-138 & Table 1: p-value?
- Line 159 & 176: Unsurprisingly, … Authors' opinions should be excluded in the Results section. Please, delete these words.
- Line 167: Given their expertise, was the SP less consistent than the GP? This is generally unacceptable. If so, please describe this in the discussion section.
- Table 2, 3, 4, 5: Please also provide the p-value difference between each group.
Discussion
- Despite some of the major concerns mentioned above, the limitation section is too simple. Please add based on this.
References
- Some of the references cited are very old. Please cite the latest references as possible. Please, check again.
Author Response
Response letter
Response to reviewer 1:
Reviewer’s general comment: This study evaluated the value of radiographic data in the diagnosis of knee osteoarthritis (OA) by general physician (GP) and secondary care physician (SP). Although this study provides new information that has not been reported so far, it has some concerns. The core information of knee OA is missing from radiographic analysis. Moreover, the statistical analysis is very poor. Therefore, it will not be able to accept this manuscript for publication in its present form. However, I think that this paper has sufficient potential to be reconsidered and evaluated again by reviewers and editors if the authors significantly alter and rework the manuscript.
Authors’ response: Thank you very much for reviewing our manuscript. It’s very meaningful to hear what other researchers think about this study, which also helps us to improve the quality of this study at the same time. For all your concerns, we prepared point-to-point responses, please see below.
The statistical analysis is really a challenging part while doing the study design. The core of the present study is to explore whether radiographs are necessary to diagnose knee OA. To answer this question, we think we should emphasize the clinical relevance rather than statistical significance. Since a large sample was included in this study, a minor difference could result in statistical significance. Therefore, we chose not to perform statistical analysis on the diagnoses and diagnostic indicators while designing this study. With your concerns in mind, we made some modifications to clarify and support our points.
Abstract
Point 1: Line 21: ‘Seventeen’… The beginning of the sentence should not start with a number.
Authors’ response: Thanks, it has been corrected.
Point 2: Line 21, 55: CHECK… The authors should define the abbreviations at their first appearance. Although it is widely accepted.
Authors’ response: Thank you for the suggestion. Since it’s in the abstract, we added a brief explanation in line 22, and a more detailed explanation was added next to its first appearance in the introduction (line 56).
Point 3: Hypothesis?
Authors’ response: Actually, we didn’t set any specific hypothesis before doing this research. Since we aimed to evaluate the added value of radiographs for diagnosing knee osteoarthritis in terms of diagnostic ability and certainty, and no such results had been reported before, we would like to set the research question as exploratory (we have added this in statistics in 137-138). Therefore, there is no hypothesis statement in this manuscript.
Methods
Point 4: IRB approval?
Authors’ response: ‘The protocol has been approved by the Ethical Committee of UMC Utrecht (protocol number 02/017-E)’. The above statement has been added at the beginning of the method section (line 63-64).
Point 5: Study period?
Authors’ response: Yes, we forget to clarify this issue in the method section. The CHECK cohort recruited patients between October 2002 and September 2005, and patients were required to be followed for 10 years. Expert diagnosis was obtained between June 2018 and January 2019.
We have added the statement about the timing of recruiting participants of CHECK cohort in line 71-73. Besides, we have stated the period of obtaining expert diagnosis in line 94-95.
Point 6: Line 74: In general, BMI is also included in the demographics. Is there any reason to separate this?
Authors’ response: Thanks for the suggestion, you are right, we have moved the BMI into demographics.
Point 7: Line 84-85: This is the major concern of this study. Did the authors only evaluate the standing AP/LAT view? It is very important to evaluate the posterior femoro-tibial joint or patello-femoral joint in radiographic diagnosis of the knee OA. Didn't the authors assess the Rosenberg view or the Merchant view? I think this must be evaluated.
Authors’ response: The AP view of radiographic assessment of knee in CHECK cohort is weight-bearing posterior-anterior fixed flexion view (see the protocol[1]). Therefore, the Rosenberg view (posterior-anterior fixed flexion view) was assessed. Maybe this confusion was caused by our unclear statements, modification has been made in line 85 and 88.
However, we didn’t provide the Merchant view for assessing patellofemoral joint, though the skyline view was taken for knees in CHECK cohort. This could be a limitation for this study, but we think its influence on our conclusion is limited. There are three main reasons: 1) as described in the main study report[1], the prevalence of patellofemoral joint OA in CHECK cohort was quite low (4.6%); 2) and its presence would also have been suggested by the lateral radiograph findings as well as clinical history and physical examination, e.g. knee crepitus [2][3].
Nevertheless, we have added this point as limitation (line 318-323).
- Wesseling, J.; Boers, M.; Viergever, M.A.; Hilberdink, W.K.; Lafeber, F.P.; Dekker, J.; Bijlsma, J.W. Cohort profile: Cohort hip and cohort knee (check) study. Int J Epidemiol 2016, 45, 36-44, doi:10.1093/ije/dyu177.
- Lankhorst, N.E.; Damen, J.; Oei, E.H.; Verhaar, J.A.N.; Kloppenburg, M.; Bierma-Zeinstra, S.M.A.; van Middelkoop, M. Incidence, prevalence, natural course and prognosis of patellofemoral osteoarthritis: The cohort hip and cohort knee study. Osteoarthritis Cartilage 2017, 25, 647-653, doi:10.1016/j.joca.2016.12.006.
- van Middelkoop, M.; Bennell, K.L.; Callaghan, M.J.; Collins, N.J.; Conaghan, P.G.; Crossley, K.M.; Eijkenboom, J.; van der Heijden, R.A.; Hinman, R.S.; Hunter, D.J., et al. International patellofemoral osteoarthritis consortium: Consensus statement on the diagnosis, burden, outcome measures, prognosis, risk factors and treatment. Semin Arthritis Rheum 2018, 47, 666-675, doi:10.1016/j.semarthrit.2017.09.009.
Point 8: Line 96-102: Is the score in units of 1 point?
Authors’ response: Yes, all the certainty scores are integers with unit of 1 point. To clarify this issue, we have added a brief statement in the parenthesis in line 101.
Point 9: Statistics: Which statistical program did the authors use? What was the significant ranges of p-value?
Authors’ response: We performed all analysis in software SPSS version 25.0 (IBM, Chicago, USA), and set the significance level at 0.05 using a 2-sided P value for all tests. This statement has been added in line 147.
Results
Point 10: To evaluate radiographic accuracy between GP and SP, it can be interesting to compare the evaluation results for K-L grade. Please also add the judgment result of K-L grade between each clinician to the result.
Authors’ response: We really appreciate this suggestion. Indeed, it could be interesting to compare the judgment result of K-L grade between them. However, in this study, as we stated in the method section, we provided radiographic data consisted of scores by trained readers evaluating standard radiographic films. The scores included the information of K-L grade (it can be seen in appendix table 1). Therefore, we didn’t ask experts to judge K-L grade by themselves, though we provided real films. By doing this, as we stated in the discussion, we believe it could be helpful to diminish the bias of image reading skill differences between individuals. Furthermore, experts were asked to make diagnosis on ‘clinically relevant OA’, which was assumed to be different from the radiographic OA (KL≥2). Our post-hoc analysis confirmed our assumption that the overlap between expert-based OA and radiographic OA was only 59% (data not shown). That’s why we didn’t emphasize the KL grade judgement in this study.
Providing the standard KL grade is not the case in the scenario of real practice, which might cause inconsistency between study results and real situations. So we added this concern as limitation (line 315-319). Even so, most clinical diagnoses remained same after viewing both the scores and films. This lends support our conclusion that radiographs seem to be redundant in most cases.
Point 11: Line 137-138 & Table 1: p-value?
Authors’ response: We have reframed the table 1 and added statistical tests for comparing characteristics between GP and SP. Some relative statements have been added in statistics section (line 124-125) and results section (line 155, 156).
Point 12: Line 159 & 176: Unsurprisingly, … Authors' opinions should be excluded in the Results section. Please, delete these words.
Authors’ response: Yes, we agree, such words have been deleted, and the sentences have been modified.
Point 13: Line 167: Given their expertise, was the SP less consistent than the GP? This is generally unacceptable. If so, please describe this in the discussion section.
Authors’ response: According to figure 2 and table 2 we could see SP were slightly less consistent than GP (consistent rate 82% vs 86%), which could be explained by the expert characteristics as SP place more emphasis on radiographs.
But when taking a look at the rate of correct amendment, we found a similar rate (9% vs 8%) between them, which means about half of the amendments caused by viewing radiographic data were incorrect. The added value of radiographs could only be reflected in these correctly amended diagnoses. Furthermore, there is no obvious difference among diagnostic indicators between GP and SP either before or after viewing radiographic data. Similar results were also found in certainty scores. Therefore, we believe the added value of radiographs should be considered as similar for GP and SP.
We have added this part in the discussion (line 296-303).
Point 14: Table 2, 3, 4, 5: Please also provide the p-value difference between each group.
Authors’ response: We have added chi-square tests for comparing overall diagnoses before and after viewing radiographic data in GP and SP. The statements were inserted in statistics part (line 123-124) and results (line164-165, 181-182). Both P values were less than 0.001 and were stated at the footnote of table 2.
As for table 3-5, we don’t think it’s very necessary to perform statistical analysis on the diagnostic indicators and in subgroups. As has been stated above, this study mainly focuses on clinical relevance rather than statistical significance (clarification has been added in line 134-138). To test the precision of estimates, we added 95%CIs for all diagnostic indicators in table 3 and 5. Generally, all the confidence intervals are relatively ‘narrow’, so it’s reasonable to use the estimated values for comparisons.
Discussion
Point 15: Despite some of the major concerns mentioned above, the limitation section is too simple. Please add based on this.
Authors’ response: Thank you for all the above comments and suggestions. We have added several limitation points according to the above comments and responses (line 316-326).
References
Point 16: Some of the references cited are very old. Please cite the latest references as possible. Please, check again.
Authors’ response: Two old citations have been replaced, and 2 new citations have been added (line 384-386, 410-411, 416-426).
Reviewer 2 Report
Authors claimed about the value of radiographs in the diagnosis of osteoarthritis of the knee. However, it is questionable whether a conclusion about diagnostic ability can be made by analyzing "the follow-up data of patients with suspected early knee osteoarthritis". Nevertheless, I think that this is a valuable study that derives interesting data using vast amounts of data that have been analyzed over a long period of time.
Words that express the degree of ambiguity such as "somewhat" and "fairly" stand out, but given the nature of this study, I think they are within acceptable limits.
1. At what point was the diagnosis made with the data collected at the time of 5, 8, and 10 year follow-up?
2. It was very interesting because it was a rare design, but I think there is a possibility that a bias caused by authority was attributable to the process of meeting and consensus between GPs and SPs from different careers.
There are also some minor questions
1. A description of the CHECK cohort study is given in the second occurrence of this word rather than the first.
2. Is it an official name to express WOMAC score as WOMAC pain, function and stiffness score?
Author Response
Response letter
Response to reviewer 2:
Reviewer’s general comments: Authors claimed about the value of radiographs in the diagnosis of osteoarthritis of the knee. However, it is questionable whether a conclusion about diagnostic ability can be made by analysing "the follow-up data of patients with suspected early knee osteoarthritis". Nevertheless, I think that this is a valuable study that derives interesting data using vast amounts of data that have been analyzed over a long period of time.
Words that express the degree of ambiguity such as "somewhat" and "fairly" stand out, but given the nature of this study, I think they are within acceptable limits.
Authors’ response: Thank you very much for reviewing our manuscript and the positive feedbacks! It’s true that making diagnosis by viewing patients’ longitudinal data is different from the scenario of real practice. Most of time, physicians don’t spend so much time for getting all the data that we provided in this study, they just ask several questions and do some brief examinations to get some ‘key’ information. Similarly, in this study, we don’t believe experts have taken all the data into consideration. Further study is needed for exploring which features (e.g. medical history, symptoms, physical examinations) are related to the diagnosis.
For all your concerns, please see our point-to-point responses below.
Point 1: At what point was the diagnosis made with the data collected at the time of 5, 8, and 10 year follow-up?
Authors’ response: The diagnosis was made after all the data of 5, 8, and 10 year follow-up were collected. The CHECK cohort recruited patients between October 2002 and September 2005, and patients were required to be followed for 10 years. Expert diagnosis was obtained between June 2018 and January 2019. Such statements have been added in the method section (line 72-73, 94-95).
Point 2: It was very interesting because it was a rare design, but I think there is a possibility that a bias caused by authority was attributable to the process of meeting and consensus between GPs and SPs from different careers.
Authors’ response: Yes, we agree this could be a bias for this study, since SPs likely have more authority than GPs, and also situations can be vary from orthopaedic surgeons vs. GP to rheumatologists vs. GP et al. In this case, the final diagnosis would be closer to SPs’ independent diagnosis, which means diagnostic indicators of SPs would be higher than that of GPs. However, according to table 3 and 5, the diagnostic indicators were quite similar between GP and SP both before and after viewing radiographic data. Additionally, we performed a very preliminary subgroup analysis by splitting SPs according to their careers, no obvious difference was found among diagnostic indicators between orthopaedists and rheumatologists (analysis is not feasible for sports physicians since only 2 sports physicians were recruited). Therefore, we don’t think it’s a big issue, but should be stated in the limitation (line 323-326).
Minor points:
Point 3: A description of the CHECK cohort study is given in the second occurrence of this word rather than the first.
Authors’ response: We have added a brief explanation of CHECK in abstract (line 22) and a detailed one next to its first appearance in the introduction (line 56).
Point 4: Is it an official name to express WOMAC score as WOMAC pain, function and stiffness score?
Authors’ response: Though it’s widely used in the literature, officially, it should be stated as ‘WOMAC subscales of pain, function and stiffness’, so we have changed the term accordingly.